# TRICIN: A Phase II Trial on the Efficacy of Topical TRIchloroacetic Acid in Patients with Cervical Intraepithelial Neoplasia

**DOI:** 10.3390/cancers14235991

**Published:** 2022-12-05

**Authors:** Richard Schwameis, Julia Ganhoer-Schimboeck, Victoria Laudia Hadjari, Lukas Hefler, Birgit Bergmeister, Tatjana Küssel, Gunda Gittler, Theodora Steindl-Schoenhuber, Christoph Grimm

**Affiliations:** 1Department of General Gynecology and Gynecologic Oncology, Gynecologic Cancer Unit, Comprehensive Cancer Center, Medical University of Vienna, 1080 Vienna, Austria; 2Department of Obstetrics and Gynecology, Konventhospital Barmherzige Brueder and Ordensklinikum Linz, 4010 Linz, Austria; 3Karl Landsteiner Institut Fuer Gynaekologische Chirurgie und Onkologie, 4020 Linz, Austria; 4Barmherzige Brueder Linz, Hospital Pharmacy, 4020 Linz, Austria

**Keywords:** cancer prevention, CIN, large loop excision of the transformation zone, conization, HPV, human papilloma virus, LSIL, squamous cell intraepithelial lesion, HSIL, TCA

## Abstract

**Simple Summary:**

Roughly, 5% of all women undergoing screening for cervical cancer suffer from cervical intraepithelial neoplasia (CIN). Persistent and high grade CIN are usually treated by conization. Conization is associated to several long-term complications such as preterm deliveries. Therefore, non-surgical treatment approaches to CIN are urgently needed. Up to now only retrospective data were available for the topical treatment with trichloroacetic acid (TCA). This is the first prospective trial showing high efficacy of TCA in the treatment of CIN. In the future, TCA may represent an effective and feasible non-surgical treatment approach for CIN.

**Abstract:**

Data on non-surgical treatment approaching persistent cervical intraepithelial neoplasia (CIN) are scarce. Retrospective analysis suggest high efficacy of topical treatment with trichloroacetic acid (TCA). This prospective phase II study set out to investigate the efficacy of a single application of 85% TCA in the treatment of CIN I/II. Patients with CIN I/II were treated a single time with 85% TCA. After three and six months colposcopic, histologic, and HPV evaluation was performed. The primary endpoint was treatment efficacy defined as complete histologic remission six months after treatment. The secondary endpoint was HPV clearance six months after treatment. A total of 102 patients with CIN I/II were included into this trial. Complete histologic remission rates were 75.5% and 78.4% three and six months after TCA treatment, respectively. Clearance rates of HPV 16, 18 and other high risk types were 76.5%, 91.7%, 68.7% after six months, respectively. Side effects of TCA were mild and lasted usually less than 30 min. This is the first prospective trial reporting high histologic complete remission rates in patients with CIN I/II after a single 85% TCA treatment. In the future, TCA may represent an effective and feasible non-surgical treatment approach for CIN.

## 1. Introduction

Cervical intraepithelial neoplasia (CIN) is the major precursor in the development of cervical cancer. Since decades the incidence of CIN is rising. In the United States, the current estimated incidence of CIN I and CIN II/III is 4% and 5% in women undergoing cervical cancer screening, respectively [1].

While women with low grade lesions (LSIL, CIN I) usually undergo observational visits, in high grade lesions (HSIL, CIN II/III) excisional procedures are generally recommended [2]. Treatment of CIN with excisional procedures is associated to several long-term side effects including an increased risk of preterm delivery [3]. This is especially concerning as the peak incidence of CIN II/III is found in women between 25 and 35 years of age [4]. Furthermore, the median age of women at first delivery is constantly rising, and therefore women with CIN II/III are frequently confronted with an incomplete family planning [5]. Hence, an effective conservative treatment of CIN without subsequent increased risk of preterm delivery is needed.

Similar to the development of CIN, the development of anal intraepithelial lesions (AIN) is strongly associated with a persistent infection with human papillomavirus (HPV) [6]. Based on two retrospective analyses, the topical use of 80–85% trichloroacetic acid (TCA) is established as a reasonable treatment approach in high grade AIN [7,8].

TCA is an analogue of acetic acid in which the three hydrogen atoms of the methyl group are replaced by chlorine atoms. It is widely used in biochemistry for the precipitation of macromolecules, such as proteins, DNA, and RNA. Solutions containing TCA as an ingredient are used for cosmetic treatments, such as chemical peels, tattoo removal, and the treatment of warts, including genital warts. Interestingly, TCA is even considered safe for treatment of warts during pregnancy [9]. Furthermore, topical TCA is one of the accepted cytotoxic therapies for condyloma and is recommended by the Center for Disease Control as one of the first-line treatments for genital warts [10].

Recently, a retrospective analysis investigating the efficacy of TCA in the treatment of CIN showed promising results for histologic regression and HPV clearance [11]. Unfortunately, these results were limited due to the retrospective design of the study, inconsistent endpoint assessment (by PAP smear or biopsy) and substantial number of missing follow-up information (e.g., information on HPV clearance after treatment). Therefore, this prospective, single arm, phase II study is set out to investigate the efficacy of a single application of 85% TCA in the treatment of patients with CIN I/II.

## 2. Materials and Methods

The **TRICIN** trial (**TRI**chloroacetic acid in patients with **C**ervical **I**ntraepithelial **N**eoplasia) was conducted at the Department of Obstetrics and Gynecology, Konventhospital Barmherzige Brueder and Ordensklinikum Linz, Austria. In Austria, there is an opportunistic screening program for CIN based on cytologic an HPV testing. Women with pathologic cytologic results were referred to our clinic for further treatment. Eligible patients aged between 18 and 50 years with a documented high-risk HPV infection, histologically proven CIN I or II, diagnosed during a satisfactory colposcopy (i.e., fully visible transformation zone and fully visible lesion), and a negative pregnancy test were included into this trial. Patients were excluded from study if they had histologically proven CIN III, inadequate colposcopy, a suspicious lesion that continued into the endocervical canal, or a negative HPV test. All eligible patients were informed about the treatment standard and were asked to participate in the present trial. All included participants provided written informed consent before study inclusion. Figure 1 shows the corresponding CONSORT diagram.

We conducted a prospective, non-randomized, single arm phase II trial (Clinicaltrials.gov Identifier: NCT04400578). The study was conducted according to the declaration of Helsinki and the ICH Harmonized Tripartite Guideline for Good Clinical Practice. Prior to initiation of the study, the protocol and all documents were approved by the ethics committee of Ordensklinikum Linz and Barmherzige Brueder Hospital Linz (48/19) and the Austrian public health agency (Österreichische Agentur für Gesundheit und Ernährungssicherheit, AGES). The AGES classified TCA as a medical device, therefore this study was conducted according to the requirements of the Austrian Medicinal Devices Act (BASG Ref. No. INS-12565011).

Eligible patients, who agreed to participate were treated topically with 85% TCA.

On the day of the inclusion visit, a standard gynecologic examination including cytology, a type-specific HPV test, a colposcopy with 3% acidic acid were performed. A colposcopically guided biopsy was taken from any visible suspicious lesion. In addition, the medical history of the patient was retrieved.

At the treatment visit, 85% TCA was applied topically to the transformation zone of the cervix in general and the cervical lesion in particular.

TCA was applied guided by colposcopy using a cotton swab. A thin film of TCA was applied to cover the complete transformation zone and any visible lesion that protruded from the transformation zone to the ectocervix. The epithelium was allowed to turn white, indicating precipitation of denatured proteins (Figure 2). Local side effects, including erythema, erosion, ulceration, or edema of the cervix/transformation zone were observed for 10 min. Participants were questioned about uncomfortable sensations using a visual analog scale from 0 to 10 (whereby the value 0 is defined as no symptom and 10 worst symptom). Uncomfortable sensations included burning or itching sensation, soreness, local pain, and vaginal discharge. Participants were advised to refrain from sexual intercourse, to use sanitary pads rather than tampons, and to shower rather than to take baths for 2 weeks. Afterwards, we handed out a diary to the participants to record side effects. Three months after treatment, participants were seen for the first follow-up visit comprising of a cytologic test, a type-specific HPV test, a colposcopy including guided biopsies. If a suspicious lesion was visible during colposcopy, at least two distinct biopsies were taken. If no lesion was visible a biopsy of each quadrant of the cervix and an endocervical curettage was taken. In addition, the patient’s diary was collected and patients were queried about adverse events, since the last visit. Adverse events asked for included lower abdominal pain, vaginal discharge or bleeding, and discomfort during sexual intercourse. Six months after TCA treatment a final follow-up visit was performed that included all examinations performed at the first follow-up visit. Two weeks afterwards, an end of study visit was scheduled, where histologic results were discussed with the patient and further treatment was assigned if necessary. At all study visits a urine pregnancy test was taken.

All examinations were performed by study investigators, experienced in colposcopy and colposcopically guided biopsy. All study investigators have a diploma in colposcopy and work regularly in the outpatient clinic for genital dysplasia. All histologic and cytologic specimens were analyzed by board-certified pathologists of the Department of Pathology of Ordensklinikum Linz which is highly specialized in gynecologic pathology.

Trichloroacetic acid (85%) was acquired from the Hospital Pharmacy of Barmherzige Brueder Hospital Linz, where it was produced on commission of Ordensklinikum Linz according to the ABO (Apothekenbetriebsordnung) in regard to the Medical Devices Regulation. At the study site, TCA was stored in 5 ml vials at room temperature. As precaution against diffusion and contamination, each vial was only used once for a single treatment.

### Statistics

The primary endpoint of this trial was treatment efficacy, defined as complete histologic remission (from any grade of CIN to no CIN) after 6 months. Secondary endpoints included histologic regression (from CIN II to CIN I), HPV clearance, and treatment-associated adverse events. Several studies suggest complete remission rates of CIN I/II without treatment of roughly 45–55% [12,13,14]. Based on a retrospective analysis of patients with CIN I to III treated with TCA a single time, a remission rate of 75–80% was expected [11]. A conservative sample size calculation revealed that a sample size of 102 patients should suffice to show a difference of 15% in treatment effect with a power of 90%.

Histologic regression, complete remission, and type-specific HPV clearance at 3 and 6 months after TCA treatment are described as proportions, and 95% confidence intervals are given. Treatment-associated side effects are given as median (range).

All analyses are based on an intention-to-treat principle. Therefore, all participants included into the trial are incorporated in the statistical analysis. Participants with missing data on histologic results and HPV clearance due to refusal to continue study participation or any other cause, were regarded as non-responders. Statistical analysis was performed using SPSS 27.0 for MAC (IBM Inc., Armonk, NY, USA). *p*-values below 0.05 were considered significant.

## 3. Results

A total of 102 patients with a median (range) age of 26.6 (19.3–50.0) were included into this trial. Table 1 shows patient demographics. At inclusion, 87 women (85.3%) had a pathologic cytologic result (34 ASC-US or ASC-H, 52 LSIL, 1 HSIL). Seventy-six (74.5%) and 26 (25.5%) of 102 patients suffered from CIN I and CIN II, respectively. In patients with CIN I 20 (26.3%), 12 (15.8%), and 63 (82.9%) patients had an infection with HPV 16, HPV 18, and other high-risk HPV types, respectively. Similarly, 14 (53.8%), 0 (0%), and 20 (76.9%) of cases with CIN II had HPV 16, HPV 18, and other high-risk HPV types, respectively (Table 2).

Following TCA treatment, a total of six patients had to be excluded from the study, due to (a) progression to CIN III (3 patients), (b) development of pregnancy (2 patients), (c) lost to follow-up (1 patients). According to the protocol, all excluded patients were included into the analyses and considered non-responders.

Three and six months after a single treatment with TCA, we assessed the histologic regression and remission rates and HPV clearance rates. After three and six months, the complete histologic remission rate in the whole group was 75.5% (66.0–83.5%) and 78.4% (95%-CI 69.2–86.0), respectively. A complete histologic remission rate of 78.4% six months after treatment is significantly higher than the anticipated spontaneous regression rate of 55% (*p* < 0.001) [12,13,14]. In the subgroups of patients with CIN I and CIN II the complete remission rates were 78.9% (95%-CI 68.1–87.5%) and 76.9% (95%-CI 56.4–91.0%), respectively (Table 3). Histologic regression (from CIN II to CIN I) was observed in 19.2% of the cohort (Table 3). Neither tobacco use, nor patients’ age had a significant impact on the efficacy of TCA (complete remission rates: 74.4% vs. 81.0% for smokers and non-smokers, *p* = 0.463; 78.5% vs. 78.4% for patients < and ≥30 years of age *p* = 0.99). Six month after TCA treatment, 70 of 102 women (68.6%) showed a normal cytologic result.

In total, 3 of 102 participants (2.9%) showed a progression to CIN III at three months after TCA treatment. Two of these patients had a CIN I and one patient had a CIN II at the time of inclusion. After TCA treatment failure, all patients underwent large loop excision of the transformation zone. The final histologic report confirmed CIN III in all three cases. A progression from CIN to invasive cancer was not observed.

Clearance rates for HPV 16, 18, and other high-risk types were 76.5%, 91.7%, and 68.7%, at six months after TCA treatment in patients with CIN I and CIN II. In the subgroup of patients with CIN II, clearance rates were 78.6%, 75% for HPV 16 and other types, respectively. No patient of the CIN II group had an infection with HPV 18 (Table 4).

Side effects were measured by using a visual analog scale. During TCA treatment, participants reported discomfort in 36 cases. Side effects lasted usually less than 30 min and all side effects were reported to be of mild severity with a median score of 0.95 (0–9), 0.0 (0–3.1), 0.0 (0–5.0), 1.0 (0.0–8.0), 1.0 (0.0–8.0) for burning, itching, soreness, pain, and vaginal discharge, respectively. During the follow-up period, participants reported possibly treatment-associated side effects in 41 cases with a median duration of 5.5. days (several minutes—30 days). The reported side effects included vaginal discharge, vaginal bleeding, lower abdominal pain, and dyspareunia. The patient-reported outcomes are shown in Figure 3. No severe adverse events or suspected unexpected serious adverse reactions were observed.

## 4. Discussion

This phase II trial demonstrates substantial efficacy of topical TCA in the treatment of CIN I and II. Six months after a single treatment the majority of patients (78.4%) showed a histologic complete remission and clearance of high-risk HPV infection. A histologic complete remission rate of 78.4% is significantly higher than the spontaneous histologic remission rate of 55% observed in the literature (*p* < 0.001) [12,13,14]. This result was irrespective of the CIN grade and HPV type. Three months after TCA treatment, three patients had progressed from either CIN I or CIN II to CIN III. No case of invasive cancer was observed. Treatment with TCA was tolerated well by all participants. No serious treatment-associated side effects were observed.

The results of this trial are in line with the data of another analysis investigating TCA in the treatment of CIN. This retrospective analysis included patients with CIN I-III that underwent a single treatment of TCA [11]. Two hundred and forty-one patients were treated with TCA and treatment response was evaluated either by cytology, histology, or type-specific HPV testing eight weeks after treatment. In this analysis, 179 patients were diagnosed with a high-grade squamous cell lesion (HSIL) and 62 with a low-grade squamous cell lesion (LSIL). The histological regression rate for HSIL was 87.7% and complete remission rate was 80.3%. For patients with LSIL, complete remission rate was 82.3%. Clearance of HPV 16, 18, and the total clearance rate were 73.5% and 75.0% and 62.8%, respectively. In accordance with the results of the current study, the retrospective analysis did not report any serious adverse events associated to the treatment or any progression from dysplastic lesion to invasive cancer.

However, besides being of retrospective nature this analysis suffered from substantial methodological limitations. First, 51 patients (21% of the population) were included into the trial without proven histologic diagnosis of CIN only based on cytologic results. Second, in 11% of patients with LSIL and in 8% of patients with HSIL data on HPV testing were not available. Third, in roughly 10% of cases no biopsies were obtained after TCA treatment. Surprisingly, these patients were not removed from analysis but were deemed as treatment responders, if cytologic results returned negative. Taken together, in roughly one-third of the patients substantial information is missing, rendering the interpretation of this retrospective analysis difficult. In comparison the current prospective study performed cytologic, histologic examination, and HPV testing before and after treatment with TCA in all patients. In addition, all drop-outs were considered non-responders. Therefore, the results of the current study are substantially more reliable.

For conservative treatment of CIN most robust data are available for the Toll-like 7 receptor agonist imiquimod (IMQ). A randomized placebo-controlled phase II trial where patients with CIN II/III were treated with local IMQ over 16 weeks showed promising results i.e., a histologic complete remission rate of 47% and a HPV clearance rate of 60% [13]. Subsequently two randomized phase III trials comparing local IMQ to conization in patients with CIN II/III were conducted. One of the two studies was prematurely terminated due to slow patient recruitment and a high rate of side effects [15]. The second study showed a significantly inferior complete remission rate of 51.9% after IMQ to conization (92.3%). The low complete remission rate was owed to the fact that in roughly 50% of patients severe adverse effects occurred which lead to treatment modification or stoppage [16]. The current trial shows substantially higher HPV clearance and histologic complete remission rates as well as a significantly lower adverse effect rate after treatment with TCA than with IMQ.

From a pathogenetic point of view, AIN is a condition very similar to CIN, as both originate from persistent high-risk HPV infections and represent precursors to invasive cancer. The risk of AIN and anal cancer is particularly high in HIV positive men who have sex with men (HIV-MSM). Recently, a retrospective analysis compared TCA against electrocautery ablation (ECA) in 230 HIV-MSM with high grade AIN [17]. Interestingly, treatment with TCA yielded significantly higher complete remission rates (60.7%) than ECA (28.0%). While these results are similar, the rate of complete remission in the current trial is substantially higher than the rates shown in the AIN study. This could be due to the fact that the patients included in the AIN trial were HIV positive, which delineates an immune incompetent state or due to the fact that remission rates after treatment for AIN are generally lower than after treatment for CIN.

Interestingly, studies performed on patients with AIN used TCA up to three times [7]. Therefore, one could also discuss whether TCA can be used multiple times at the cervix to treat CIN. However, at the time the current study was planned, no data on multiple application of TCA to the cervix were available. Interestingly, meanwhile a small study investigated multiple TCA application in patients with CIN 1–3 [18]. This study was based on a rather unclear sample size calculation and did not show any impact of multiple TCA application on the regression rate of CIN. However, as stated above, the statistical design of this trial is unclear, therefore the results have to be considered with utmost caution.

The present study showed substantial effect of TCA in the treatment of patients with CIN I and CIN II. TCA denaturizes proteins and in high concentrations leads to cell death [19]. TCA could serve as a possible treatment option in patients with CIN I and CIN II, before progression to CIN III occurs.

Indeed, additional studies should be conducted before using TCA in the treatment of CIN I and II on a daily basis. Moreover, the most intriguing question remains the efficacy of TCA treatment in patients with HSIL/CIN III. In the majority of cases, patients with HSIL (CIN III) are subjected to excisional procedures such as conization. Therefore, the most reasonable implication of the outcome at hand is to compare TCA treatment with excisional procedures such as large loop excision of the transformation zone in patients with HSIL/CIN III in a prospective trial. From an academic point of view, it is crucial to conduct such a study before introduction of TCA treatment into daily clinical routine.

This study is an important addition to the clinical knowledge about TCA treatment in patients with CIN due to its rigorous prospective design and statistical power to assess the efficacy of TCA. Compared to other published studies, the prospective nature and the integrity of the data have to be highlighted. Afterall, this study also has its limitations. Most importantly, this study did not include patients with CIN III, the population with the highest need for treatment. However, as there is a risk of 12–40% of progression to invasive cancer [20,21] and this was the first prospective trial that investigated the efficacy of TCA in CIN, it would have been unethical to include these patients into the on-hand trial. Another limitation is the relatively short follow-up period of only 6 months. However, we chose six months as end of study time point as data suggest that results taken six months after excisional procedures are most reliable to assess the treatment result [22].

## 5. Conclusions

In conclusion, we present the first prospective trial showing high histologic complete remission rates in patients with CIN I and CIN II after a single TCA treatment. Honestly, the results of our trial are most promising. However, prior to introduction of TCA into daily clinical routine, the current data have to be confirmed in a prospective phase III trial including patients with CIN III.

## Figures and Tables

**Figure 1 cancers-14-05991-f001:**
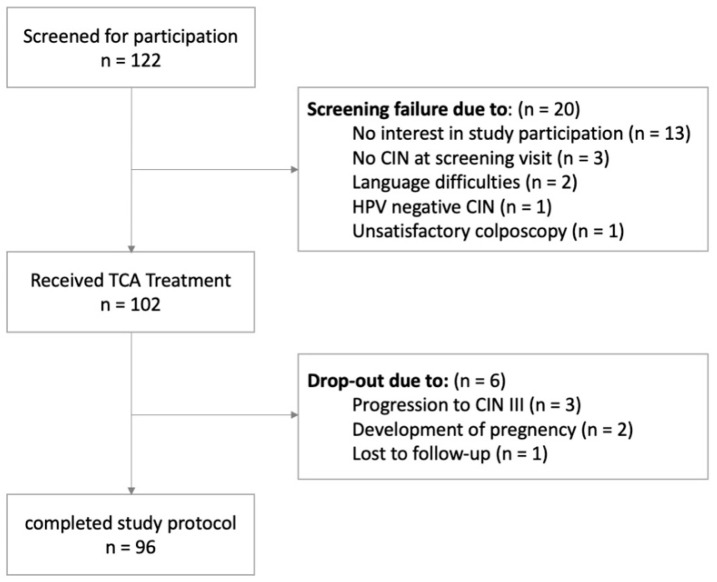
Consort diagram TCA 85% chloroacetic acid, CIN: cervical intraepithelial neoplasia. HPV human papilloma virus.

**Figure 2 cancers-14-05991-f002:**
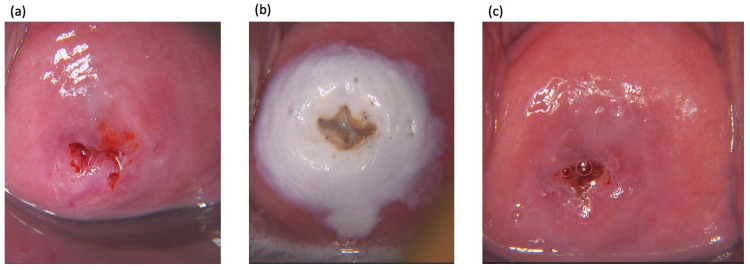
Colposcopic pictures of the cervix taken (**a**) before, (**b**) 10 min and (**c**) 3 months after local treatment with 85% trichloroacetic acid.

**Figure 3 cancers-14-05991-f003:**
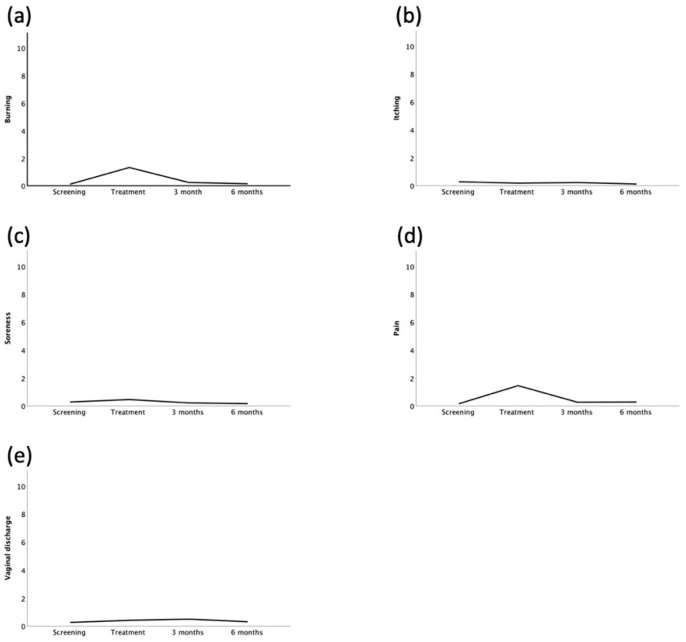
Patient-reported outcomes regarding treatment-associated symptoms before, during, 3 and 6 months after a single treatment with 85% trichloroacetic acid. (**a**) Burning, (**b**) itching, (**c**) soreness, (**d**) pain, and (**e**) vaginal discharge.

**Table 1 cancers-14-05991-t001:** Demographics of 102 patients with cervical intraepithelial neoplasia undergoing treatment with 85% trichloroacetic acid.

Variable	n (%)
**Patient’s age** *	26.6 (19.3–50.0)
**Tobacco use**	
non-smoker	63 (61.8)
<5 cigarettes/day	17 (16.7)
5–9 cigarettes/day	8 (7.8)
10–19 cigarettes/day	12 (11.8)
>20 cigarettes/day	2 (2.0)
**Body Mass Index** (kg/m^2^)	23.7 (4.8)
**Number of previous sexual partners ***	7 (1–65)
**Immunologic disorder**	
None	101 (99.0)
Crohn’s disease	1 (1.0)
**History of STD**	
HPV infection	92 (90.2)
Chlamydia infection	5 (4.9)
Previous Treatment for CIN	3 (2.9)
Condylomata accuminata	2 (2.0)
**Previous pregnancies ***	0 (0–4)
**Previous deliveries ***	0 (0–3)
**Days from last menstrual period ***	18.5 (10.0–27.75)
**Current contraception method**	
None	11 (10.8)
Oral contraceptive	45 (44.1)
Intravaginal ring	3 (2.9)
Barrier method (condom)	36 (35.3)
Cooper IUD	5 (4.9)
Hormonal IUD	1 (1.0)
Unknown	1 (1.0)

n number, BMI body mass index, STD sexually transmitted disease, HPV human papilloma virus, CIN cervical intraepithelial neoplasia, IUD intrauterine device. * given as median (range).

**Table 2 cancers-14-05991-t002:** Rates of HPV 16, 18 and other high-risk types in patients with cervical intraepithelial neoplasia grade I and II undergoing treatment with 85% trichloroacetic acid.

	CIN I (*n*/%)	CIN II (*n*/%)
**Number of patients**	76 (74.5)	26 (25.5)
**HPV 16**	20 (26.3)	14 (53.8)
**HPV 18**	12 (15.8)	0 (0)
**Other high-risk HPV**	63 (82.9)	20 (76.9)
**multiple HPV types**	17 (22.4)	8 (30.8)

CIN cervical intraepithelial neoplasia, HPV human papilloma virus.

**Table 3 cancers-14-05991-t003:** Histopathologic response rates (remission and regression) of patients with cervical intraepithelial neoplasia grade I and II 3 and 6 months after a single topical treatment with 85% trichloroacetic acid.

	3 Months (*n*/%)		6 Months (*n*/%)	
Lesion Grade at Study Inclusion	CIN I (*n* = 76)	CIN II (*n* = 26)	CIN I (*n* = 76)	CIN II (*n* = 26)
**Remission**				
No CIN	56 (73.7)	21 (80.8)	60 (78.9)	20 (76.9)
**Regression**				
CIN I	-	3 (11.5)	-	5 (19.2)
**Persistence**				
CIN II	-	1 (3.8)	-	-
CIN I	14 (18.4)	-	8 (10.5)	-
**Progression**				
CIN III	2 (2.6)	1 (3.8)	-	-
CIN II	1 (1.3)	-	3 (4.2)	-
**Drop out**	3 (3.9)		5 (6.6)	1 (3.8)

CIN cervical intraepithelial neoplasia.

**Table 4 cancers-14-05991-t004:** Human papillomavirus (HPV) clearance rates 3 and 6 months after a single topical treatment of with 85% trichloroacetic acid.

	3 Months			6 Months		
Lesion Grade	HPV Type 16	HPV Type 18	Other HR Types	HPV Type 16	HPV Type 18	Other HR Types
CIN II Clearance	71.4% (41.9–91.6%)	-	75.0% (50.9–91.3%)	78.6% (49.2–95.3)	-	75.0% (50.9–91.3%)
CIN I Clearance	85.0% (62.1–96.8%)	81.8% (48.2–97.7%)	60.3% (47.2–72.4%)	75.0% (50.9–91.3%)	91.7% (61.5–99.8%)	66.7% (53.7–78.0%)
Total Clearance	79.4% (62.1–91.3%)	81.8% (48.2–97.7%)	63.9% (52.6–74.1%)	76.5% (58.8–89.3%)	91.7% (61.5–99.8%)	68.7% (57.6–78.4%)

CIN cervical intraepithelial neoplasia, HPV human papilloma virus, HR high risk.

## Data Availability

The data presented in this study are available on request from the corresponding author.

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
