# Peer review of "TRICIN: A Phase II Trial on the Efficacy of Topical TRIchloroacetic Acid in Patients with Cervical Intraepithelial Neoplasia"

_cancers, 2022, doi:10.3390/cancers14235991_

Round 1

Reviewer 1 Report

The reviewed manuscript reported interesting results from a prospective trial examining the efficacy of topical use of 85% trichloroacetic acid (TCA) in the population of patients diagnosed with CIN 1 or 2.

The study design was planned very well. The authors described their research clearly and showed interesting results, that could encourage further studies such as comparing TCA with other non-excisional methods of CIN1/2 management (cryotherapy, laser ablation) or validating TCA topical use also in CIN 3 population in comparison with the standard of care which is conisation.

The most valuable features of this article are the prospective nature of the research (for the first time ever regarding the use topical use of TCA) and its definite outcome as an effective and feasible management that could be offered to women with persistent CIN1 or CIN2.

The are just minor remarks pointed below requiring authors clarification:

1.       In lines 102-105 authors described procedures during „inclusion” visit: examination; cytology; HPV testing and colposcopy with biopsy. It is not clear how they found such patients population – was there any pre-screening based on abnormal Pap smear? Usually, after abnormal screening test other verification procedure is performed and it takes time, especially in case of biopsy for patologic assessment. How come the authors did those procedures in one day?

2.       During 3 month follow-up visit and final visit (6 month) the authors performed cytology, HPV testing and colposcopy with biopsy. They described very well the primary endpoint that was histologic complete regression of CIN, they also described HPV clearence but I could not find the comment to cytology results from the follow-up visits. Even if cytology normalization was not recognized as secondary endpoint it is worthy to know what was the impact of TCA treatment on cytology results.

Reviewer 2 Report

The manuscript by Schwameis et al. reported a phase II trial on the efficacy of topical trichloroacetic acid in patients with cervical intraepithelial neoplasia. The results showed hihh histologic complete remission rates in CIN I/II patients after a single TCA treatment, with no severe side effects observed in 6 months. As the first prospective clinical trial, this study could be of significant importance for introducting TCA as an effective non-surgical clinical approach for CIN. I have the following comments for the manuscript.

Although the word TRICIN was shown in the title, it has not been mentioned or explained in the manuscript. It could be confusing between tricin, another chemical, with TCA.

Line 113: range of scale needs to be indicated.

Table 1 showed the Tobacco use of the patients, but the authors did not discuss the correlation of tobacco use and the outcome of the treatment. Also, does age affect the treatment of TCA?

Line 184-185: reference for the spontaneous regression rate not cited.

Line 249: compression should be comparison.

As the application of TCA is non-surgical and relatively convenient, could multiple applications of TCA increase the efficacy? What is the reason to choose one application as the treatment for the current study?
